# Active Inference for Learning and Development in Embodied Neuromorphic Agents

**DOI:** 10.3390/e26070582

**Published:** 2024-07-09

**Authors:** Sarah Hamburg, Alejandro Jimenez Rodriguez, Aung Htet, Alessandro Di Nuovo

**Affiliations:** Department of Computing, Sheffield Hallam University, Sheffield S1 1WB, UK; a.jimenez-rodriguez@shu.ac.uk (A.J.R.); aung.htet@student.shu.ac.uk (A.H.); a.dinuovo@shu.ac.uk (A.D.N.)

**Keywords:** active inference, neurorobotics, developmental robotics

## Abstract

Taking inspiration from humans can help catalyse embodied AI solutions for important real-world applications. Current human-inspired tools include neuromorphic systems and the developmental approach to learning. However, this developmental neurorobotics approach is currently lacking important frameworks for human-like computation and learning. We propose that human-like computation is inherently embodied, with its interface to the world being neuromorphic, and its learning processes operating across different timescales. These constraints necessitate a unified framework: active inference, underpinned by the free energy principle (FEP). Herein, we describe theoretical and empirical support for leveraging this framework in embodied neuromorphic agents with autonomous mental development. We additionally outline current implementation approaches (including toolboxes) and challenges, and we provide suggestions for next steps to catalyse this important field.

## 1. Embodied Agents: Inspiration from Humans

One major limitation of mainstream machine learning is the absence of a body to sup-port self-determined learning via autonomous interaction with the environment [1]. This limitation is intrinsic to technologies like neural networks because they are inspired only by the neural components of the complex interacting subsystem that constitute a living being. Embodied agents, on the other hand, have a physical implementation (such as a body, drone, or vehicle), and exist on a continuum from weak to strong forms of embodiment (see [2] for discussion).

Neuropsychology research has extensively evidenced the embodied nature of human cognitive abilities. These are formed not only within the brain but also shaped by the body and the experiences acquired through it, both through actions, such as manipulatives, gestures, and movements [3,4,5], and through sensations, such as touch [6] and odours. Even purely mental phenomena like mental simulation are shaped by embodied experiences and, in turn, shape different aspects of human behaviour [7].

Importantly, the human brain is subject to energy constraints and is considered highly energy efficient (estimated power consumption is 20 Watts per day [8]—the equivalent of one energy-efficient lightbulb). For context, the current fastest supercomputer in Europe has been described as “exceptionally green” and consumes 8.5 million Watts per day [9]. While supercomputers consume multiple orders of magnitude more energy than the human brain, it is precisely the energy constraints of the latter that give rise to many of its computational properties and, ultimately, to the cognitive abilities that make us human [10].

Cognitive domains that humans excel in include learning with minimal data (aided via predictions based on mental imagery models), abstract thinking (understanding concepts that are not immediately present or tangible), creativity, problem solving, adaptability, reasoning, decision making, and collaboration (e.g., [11,12,13,14]). Importantly, these capacities are enabled by the relatively large amount of data a human is exposed to throughout development, on one hand, combined with extensive real-time multimodal sensory input (i.e., through the skin). These constitute two novel scaling frontiers yet to be exploited by any learning algorithm to date.

Inspired by the brain, neuromorphic computing aims to create powerful energy-efficient cognitive systems and has been described as a “key enabling technology for the development of a unique generation of autonomous agents” [15]. Neuromorphic systems include plastic “synapses” (i.e., mediated by spike-timing-dependent plasticity (STDP)), and dynamic and “event-based” computing paradigms (for a review, see [16,17]). Efficiency in this context is relative to the particular energy constraints imposed by the environment and the body itself.

For instance, synapses are dynamic elements that are up- or down-scaled (i.e., pruned) depending upon said energy constraints. Extensive synaptic pruning is a crucial neurodevelopmental process whereby about 50% of the brain’s synapses are removed throughout early childhood into adulthood—enhancing energy efficiency [18]. Myelination is also a fundamental neurodevelopmental process, whereby the nervous system becomes increasingly energy efficient through the “insulation” of specific neurons at critical timepoints during development, in a coordinated pattern aligned with developmental requirements.

The dynamic aspect of computation allows for continual learning and adaptation and facilitates the agent’s coupling to environmental rhythms. In particular, the temporal architecture of the brain becomes more synchronised over neurodevelopment [19]. Finally, the event-based nature of computation allows for the agent to be acted upon by the environment (for example, it is very difficult to avoid an odour).

### Tackling Developmental and Perceptual Scaling Limits

Research from developmental sciences has shown that children acquire cognitive skills—including perceptual [20], language [21], social [22], and numerical [23]—through “embodied experiences” (i.e., motor movements and interactions with objects and people). Emulating these embodied learning mechanisms in artificial agents may, therefore, promote the acquisition of human-like abilities. This developmental approach aligns with the 1950 hypothesis by Alan Turing, “Instead of trying to produce a program to simulate the adult mind, why not rather try to produce one which simulates the child’s? If this were then subjected to an appropriate course of education one would obtain the adult brain” [24].

How best to enable embodied agents to “learn like a human” remains highly conceptual. The combination of developmental robotics and neuromorphic computing is leading to the emerging field of developmental neurorobotics (see section below), which aims to inform our understanding of human-like learning mechanisms for embodied neuromorphic systems. These aspects of embodied learning need to be necessarily neuromorphic due to the importance of distributed perceptual systems like the skin and the intrinsic relationship between development and neural remodelling in the brain [18]. 

We propose the active inference framework (AIF) as an important tool for implementing learning mechanisms in embodied neuromorphic agents. Originating from neuroscience, the AIF offers a biologically plausible and unified explanation for how the brain processes information, learns, and generates behaviour, involving embodied perception–action loops [25]. This includes solving “hard exploration problems” and accounting for mechanisms of natural agency and behaviour [26]. Importantly, neuromorphic computing systems are thought to be well suited for implementing AIF principles, while in turn, AIF drives the development of neuromorphic architectures [27]. Even though these connections remain only theoretical, there has been considerable progress in process theories that help bridge the gap between theory and implementation of active inference [28].

## 2. Developmental Neurorobotics

Traditional machine learning (ML) to date is not well suited to simulate continuous real-world embodied learning. In particular, traditional ML:Requires extensive training, computation, memory, and energy—impacting scalability.Does not cope well with noise, variability, and uncertainty—impacting real-world applications.Has difficulties generalising across tasks and environments.Lacks common sense—it is not able to infer, understand, or explain [29].Lacks sufficient transparency [30].Has poor performance on tasks requiring embodied intelligence [15].

This state of affairs is connected with the existing paradigms of training those systems and perhaps to the concept of training itself.

Developmental neurorobotics is an interdisciplinary research paradigm combining computational modelling, developmental psychology, and robotics to realise an embodied artificial intelligence [31]. A pervasive paradigm in this field is Piaget’s theory of cognitive development in children (1950s; [32]). In accordance with this theory, the developmental neurorobotics approach emphasises self-determined learning via interaction with the environment. Embodied agents build models (i.e., learn) based on their own interactions with the world, rather than relying on off-the-shelf pre-trained models. Robots must continuously infer, predict and adapt, and acquire new knowledge with only weak- or self-supervision (i.e., socially guided or autonomous). This helps robots to interact effectively and dynamically with their environment across a range of tasks.

It is anticipated that developmental neurorobotics systems will better serve humanlike learning—including dynamic adaptation and generalisation across tasks. In turn, such systems will also serve as new tools for investigating human development and cognition. However, the practical considerations of this paradigm remain a challenge in the field. The questions of how to set up an effective learning loop and how to exploit the different timescales of development necessitate novel approaches that extend the gradient descent algorithmic view that dominates the field.

As yet, there are no widely accepted or unified learning frameworks employed in developmental neurorobotics systems. These systems are complex, involving the integration of different modalities and learning over long timescales across different tasks. Frameworks underlying such learning may, therefore, help guide the design of these complex systems. We propose that the active inference framework (AIF) may offer such an opportunity.

## 3. Active Inference: A Promising Framework for Learning in Embodied Neuromorphic Agents

### 3.1. Active Inference

We aim here to provide a high-level overview of AIF relevant to developmental neurorobotics. For an in-depth explanation and discussion of AIF, see [33]. We additionally provide a summary box of key concepts, benefits, and challenges of AIF.

Within AIF, the brain models the world as a set of probabilities (a generative probabilistic model), which it uses to make inferences about the world and predict what is likely to happen next. The agent actively works to minimise “surprise” by bringing predicted and observed states of the world into alignment. (“Surprise” is a measure of uncertainty about the world, taking into account the quality of data. Free energy is an upper bound to surprise). Surprise minimisation is achieved by (1) adjusting models (i.e., altering perception), and/or (2) selecting actions that aim to maximise information gain and minimise prediction errors (e.g., turning your head towards an unknown noise).

AIF involves a continual loop of perception, prediction, and action. Over short timescales, perception optimises beliefs about the current state of the world (variational free energy—VFE), while over long timescales, learning optimises beliefs about the relationships between the variables that constitute the world (expected free energy—EFE) [34]. Both occur through the minimisation of “free energy” over observations in the VFE or over actions in the EFE. The ability to account for both short- and long-term learning, in addition to the hierarchical nature of the framework (i.e., deep active inference), is particularly advantageous for neurodevelopmental frameworks, which operate over and integrate these different timescales in learning.

Compared to reinforcement learning (RL), AIF offers a more integrated view of perception and action, along with more flexible goal-setting based on prior preferences. In RL, the reward function defines an agent’s goal and allows it to learn how to best act within the environment to maximise an expected reward. In AIF, any type of outcome may be more or less preferred—the implicit reward is a feature of the agent in conjunction with the environment it inhabits (for discussion, see [35]). AIF bypasses problems associated with defining reward functions (which can be difficult, particularly for real-world tasks [26]) and instead replaces these with prior beliefs about preferred outcomes. Agents are able to learn their own prior preferences and goals are flexible. In this way, AIF extends RL, encourages exploration and information seeking, and equips agents with intrinsic curiosity [36,37].
**What is Active Inference?**Active inference is a framework for understanding how systems autonomously perceive, learn, and act. Rooted in Bayesian inference, it combines elements of perception, action, and learning into a unified theory. Originating from neuroscience, it is increasingly applied in machine learning and robotics. **Key Concepts:****Bayesian Inference:** This is used to update beliefs about the world based on new sensory information. Beliefs are based on prior knowledge and likelihood. AIF performs approximate Bayesian inference.**Generative Models:** Sensory input is predicted based on internal representations of the world. The goal of active inference is to minimise a measure of the difference between predicted and actual sensory input (prediction errors).**Free Energy Principle:** This central principle posits that biological systems act to minimise a function called "free energy”.**Variational Free Energy:** VFE quantifies the difference between the system’s internal model and the actual data observed. Minimising this ensures the model accurately reflects the observed data.**Expected Free Energy:** EFE predicts free energy that will be encountered under different possible actions or action sequences. Minimising this leads to optimal decision making.**Markov Blankets:** These represent conceptual boundaries (in terms of conditional independence) that separate a system from its environment. Systems update beliefs and make predictions about their environment based solely on the information contained within the blanket.**Partially Observable Markov Decision Processes:** POMDPs provide a mathematical framework for modeling decision making with partial information in active inference. The free energy principle guides action selection and belief updating within this framework.**Action and Perception:** These phenomena are intertwined. Actions are performed to reduce the surprise with respect to generative models (i.e., to reduce prediction error). This contrasts with traditional views that separate perception (passive) and action (active).**Learning and Adaptation:** Systems continuously update their generative models to improve their predictions, allowing for adaptation to changing environments. In this case, learning is seen as inference on the parameters of the generative model.  **Key Benefits:****Unified Framework:** The unified framework integrates perception, action, and learning into a single model, enabling a unified approach to creating intelligent and self-organising systems.**Adaptive Learning:** This continuous updating of beliefs and models allows for real-time adaptation to new information.**Embodied and Situated:** Perception and cognition are intertwined with the agent’s body (embodied) within a particular environment (situated). **Key Challenges:****Complexity:** The mathematical and computational complexity of implementing active inference can be high.**Scalability:** Scaling active inference models to large and complex applications remains an ongoing challenge.

### 3.2. Key Features of Embodied Learning and Development in the Light of AIF

There is both theoretical and empirical support for AIF as a promising tool for learning and development in embodied neuromorphic agents. First, we highlight key features of embodied learning and development [32]:Learning is cumulative and progresses in complexity.Concrete and abstract concepts are a continuum; both are learned by linking concepts to embodied perceptions [38].Learning results from self-exploration with the world, often in combination with social interaction.The importance of sensorimotor skills (including the discovery of one’s own body), communication skills, and social skills.

We have additionally summarised the current literature outlining the requirements for embodied “intelligent” agents (Table 1).

Krichmar [40] stated that neurorobotics is the ideal methodology to address brain-like features and called for the community to focus on general cognition rather than particular behaviours. Additional emphasis is placed on the ability to continuously infer, predict, and adapt (e.g., [15]).

### 3.3. Theoretical Support for Active Inference in Embodied Neuromorphic Agents

Embodiment is a key feature of AIF [44,45]—perception and cognition are deeply situated and intertwined in the embedded context of the agent and its environment [46]. In AIF, the brain has even been described as “taking a back seat to the body” [46]. This is similar to the concept of morphological computation (whereby certain processes are performed by a robot’s body with minimal neural control [47]). The importance of morphological computation in neurorobotics was emphasised by Krichmar [40].

There is also an emphasis on brain–body–environment interactions. In AIF, classic physical world distinctions between “agent” and “environment” do not exist. Instead, the statistical tool of Markov blankets is employed in models to represent boundaries between systems with an external and internal state. Sometimes blankets correspond to a physical boundary (e.g., a cell exchanging energy in a tissue), while other times they do not (e.g., external force in a moving pendulum) [48]. This particular conceptual aspect fits the requirements of the computational needs required to exploit the extent of distributed sensory systems like the skin.

Furthermore, the drive to reduce uncertainty, which underpins the AIF, has been described as comparable to curiosity [34]. The emphasis on curiosity-driven embodied activity in AIF suggests it may offer a useful framework for self-supervised learning. Curiosity is widely accepted as a powerful driver of cognitive development in humans.

Da Costa et al. [49] described AIF as an interesting framework for robotic applications because it unifies state-estimation, control, and world model learning as inference processes that are solved by optimising a single objective function—free energy. Furthermore, it endows robots with adaptive capabilities, which are central to real-world applications. According to [49], key features of AIF which may address current challenges in robotics include:Accurate and robust state tracking, including the integration of multiple sensory streams.Adaptive model-based and shared control.Learning and grounding, including learning from sparse and noisy observations and organisation of knowledge into hierarchical modular representations.Operational specification, safety, transparency, and explainability.Energy efficiency, qualified by the considerations mentioned above.

Based on these properties, AIF may lead to the development of robots that are context adaptive, safe, social, and collaborative [49]. The outlined properties of AIF also confer benefits for regulated use cases (e.g., medical applications), wearables and exoskeletons with a degree of autonomy, and neurotechnologies integrated with the human nervous system [49]. Additionally, emphasising the role of principles (like the FEP) beyond individual algorithms appears to be an attractive solution for the satisfaction of abstract norms of behaviour like social norms.

Efficiency is also emphasised by the AIF. According to [25], optimising a generative model is comparable to optimisation under the efficient coding principle. Furthermore, the FEP has been shown to encompass other principles like the efficient coding hypothesis [25]. Efficiency, in this case, is a contingent aspect because even though the FEP is theoretically efficient from the mentioned points of view, current implementations still rely on gradient descent optimization and are, nevertheless, inefficient computationally. This is bound to change with emerging neuromorphic approaches and implementation.

Conceptually, it was claimed by Lanillos et al. [36] that AIF could potentially generate high-order cognitive and metacognitive capabilities, such as monitoring, self explainability, and to some degree, “awareness”. The authors claimed this is because the model does not just predict sensations, but also their predictability, enabling robots to monitor their precision about prediction, and thus, become self-attentive.

Finally, both AIF and neuromorphic computing originate from neuroscience. It is claimed that the emphasis on neuroscience in AIF reduces the gap between engineering and life sciences [49] and makes it possible to bridge connections between neuroscience, robotics, AI, and psychology [34]. Thus far, much of the work in neuromorphic computing has focused on hardware development [17]. With regard to neuromorphic computing algorithms, [17] outline the clear benefits for neural network-style computation. Together, this suggests that AIF may be particularly suited to neuromorphic agents.

### 3.4. Empirical Support for Active Inference in Embodied Neuromorphic Agents

AIF has been shown to perform as well as traditional ML methods in some simple environments, and research indicates it may perform better in environments featuring volatility, ambiguity, and context sensitivity [49]. Learning has included the ability to generalise prior knowledge to new stimuli, resulting in a “one-shot learning” capacity qualitatively similar to that observed in humans [34].

Implementations of AIF in embodied systems have included simulated robot arms for searching, reaching, and manipulating [50,51,52]; a model for estimation and control in a humanoid robot [53,54]; and multisensory body perception and action in a humanoid robot [55]. 

The latter of these studies used proprioceptive and visual input and demonstrated natural behaviours for upper body reaching and head object tracking. The AIF algorithm was validated in terms of noise robustness and multisensory integration and was also applied for reaching and grasping a moving object [55]. The results show that closed-loop adaptation and multisensory integration are two important characteristics of the AIF approach [55]. 

Together, studies have demonstrated the benefits of AIF for robot body estimation [53,55], navigation [56,57], fault-tolerant behaviour [58], planning [36], self/other distinction [54], and complex social cognition [59]. AIF appears particularly useful for applications where the dynamics of the robot and/or the task are uncertain [36].

Within neuromorphic research, Isomura et al. [60] recently used a biologically plausible “canonical neural network” architecture to explore AIF. The authors cast the cost functions of these networks as variational free energy under an implicit generative model. Results suggested that the delayed modulation of Hebbian plasticity (a process integral to forming associations between neurons in delayed reward tasks) is a realisation of active inference. The authors suggest that casting cost functions in this way may dramatically reduce the complexity of designing self-learning neuromorphic hardware by offering a simple architecture with low computational cost. Such human-like plasticity is also likely advantageous for neurodevelopmental approaches.

Gandolfi et al. [61] recently explored AIF principles in a biologically plausible neuromorphic system built with conventional off-the-shelf hardware (low-powered microcontrollers). The authors created a simplified cerebellar circuit simulating a delayed eyeblink classical conditioning paradigm. In this model, neurons and synapses adjusted their activity to minimise their prediction error. This led to associative plasticity and unsupervised learning. Importantly, inference capabilities emerged entirely from connectivity and from the “Bayesian engine” in neurons (whereby individual neurons acted to minimise their prediction error) rather than from hardwired learning rules. Learning was also particularly rapid and required considerably fewer units and less power than traditional bioinspired networks.

Gandolfi et al. [61] highlight the remarkable simplicity of the generative model entailed by each element—each assumes one hidden state that can be in one of two levels. This is likened to spin-glass models in physics, in which the states can be “up” or “down”. Each element infers whether its world is in an “up” or “down” state by assimilating sensory (presynaptic) input. The authors suggest their experiments could be adopted to implement brain-like predictive capabilities into neuromorphic computers and robotic controllers.

AIF was also recently explored in a novel biosynthetic neuromorphic system called “DishBrain”. In this system, in vitro neural networks were integrated with in silico computing and embedded in a simulated-game world mimicking “Pong” [62]. The system was closed-loop in that it provided feedback on the causal effect of its behaviour, which afforded the in vitro culture “embodiment”. Note, the authors use the term “embodiment” here to refer to the “separation of internal versus external states where feedback of the effect of an action on a given environment is available” [62] (this aligns with functional perspectives of embodiment as discussed in [2]).

Applying AIF principles resulted in apparent learning within 5 min of real-time gameplay, and the system was said to exhibit “synthetic biological intelligence”.

As yet, we are not aware of any studies implementing AIF in neurorobotic systems (robots using neuromorphic hardware). We have, however, identified two recent studies using Bayes frameworks with neuromorphic hardware for robotic navigation and vision. Bayesian and AIF models share some similarities in terms of their probabilistic modelling approach. However, Bayesian models typically treat action as a separate stage of processing (decisions are made based on inferred probabilities), while AIF models treat action as an integral part of perception (the brain actively seeks out sensory information that reduces uncertainty and prediction error).

Tang et al. [63] proposed a spiking neural network (SNN) architecture with Bayes inference to solve a crucial problem in robotics—simultaneous localization and mapping (SLAM). The authors integrated this neuromorphic algorithm into neuromorphic hardware (Intel’s Loihi processor) and demonstrated 100 times less energy and comparable accuracy to the widely used GMapping algorithm. Ayyad et al. [64] presented a neuromorphic vision-based approach to robot perception and control. Their novel approach for the event-based detection and tracking of circular objects in the scene leveraged a Bayesian framework to retain the asynchronous nature of neuromorphic cameras.

Finally, it is noteworthy that AIF has been linked to development, including morphogenesis (a form of collective cellular intelligence) [65,66]. Pio-Lopez et al. [65] demonstrated that AIF could be used to simulate disorders of morphogenesis, potentially leading to new treatment approaches for developmental disorders.

We envisage the role of AIF in development may be conceptually extended through framing cognitive neurodevelopment as: (1) adjusting models (i.e., incorporating new rules), and/or (2) selecting actions that aim to maximise information gain and minimise prediction errors (i.e., neurodevelopmental processes, such as myelination and synaptic pruning). In this way, agents act on their “internal” environment to progress developmentally. This has implications for the design of developmental neurorobotic systems.

### 3.5. Summary

There is considerable theoretical support for utilising AIF in embodied neuromorphic agents. Empirical support is also emerging. The features and mechanisms discussed above demonstrate how embodied neuromorphic AIF agents have the potential to meet all the requirements for embodied “intelligence” outlined previously (see summary Table 1). Additional benefits identified include self-generated learning (through mechanisms akin to innate curiosity), meta-cognitive abilities, including self-monitoring and explainability (conferring safety benefits for socially collaborative and regulated use cases), multisensory, and low-power consumption.

According to Bartolozzi et al. [15], “A real breakthrough in the [neurorobotics] field will happen if the whole system design is based on biological computational principles, with a tight interplay between the estimation of the surroundings and the robot’s own state, and decision making, planning and action”. We argue that AIF may provide this, and potentially, in turn lead to the development of new tools for investigating cognitive neurodevelopmental processes.

## 4. Current AIF Implementation Approaches in Embodied Agents

According to de Vries [67], to implement AIF, at a minimum, engineers must express an agent’s prior preferences or constraints that underwrite behaviour. Design should focus on two tasks:Specification of the agent’s model and inference constraints.A recipe to continually minimise the free energy in that model under situated conditions driven by environmental interactions.

Current open-source tools for creating AIF agents include Statistical Parametric Mapping (SPM), PyMDP [68], ForneyLab [69] and RxInfer [67]. The latter of these uses a reactive message-passing framework whereby the inference process consists entirely of a (parallelizable) series of small steps (messages) that individually and independently contribute to free-energy minimisation [67].

As an example, the standard implementation approach in AIF research at present, for the discrete case, is the partially observable Markov decision processes (POMDP). A step-by-step guide to this is provided by [34]. In short, the POMDP formulation “describes beliefs about abstract states of the world, how they are expected to change over time, and how actions are selected to seek out preferred outcomes or rewards based on beliefs about states” [70]. Due to the concept of partial observability, an agent can be uncertain about its beliefs (states are “hidden”). It infers how likely it is to be in one hidden state or another based on observations (i.e., sensory input), which provide probabilistic information. The Markov property ensures that decision making implicitly includes all relevant knowledge about past states within current state beliefs [70]. Despite POMDP being a popular implementation approach in AIF research, robotics studies have used a range of different methods (see Table 2 for summary).

A key challenge for the AIF approach is the design of meaningful generative models and prior beliefs (preferences). Lanillos et al. [36] suggest that the required set of priors could be obtained through a proper curriculum, within a framework by which humans teach robots. This developmental approach is similar to Turing’s hypothesis outlined above.

## 5. Suggestions to Catalyse AIF for Developmental Neuromorphic Agents

### 5.1. Approaches

There are a number of potential avenues to integrate AIF for learning and development in embodied neuromorphic agents. When considering future directions, it is noteworthy that AIF is a multi-scale framework, and therefore potentially applicable across the neuromorphic “stack”—from hardware and sensors to modality integration, computations, and global cognitive orchestration.

One potential approach is to demonstrate a previous AIF robotic study in neuromorphic hardware. For example, the study by Traub et al. [74] may be particularly applicable for translation into neuromorphic systems as it utilized recurrent spiking neural networks (SNNs) and the eligibility propagation algorithm (“e-prop”; a method for training SNNs which relies on local learning rules at each synapse). This algorithm has recently been implemented in SpiNNaker [75] and SpiNNaker 2 [76]. Traub et al. [74] noted that e-prop is ideal for continuous online learning scenarios. An alternative approach would be to implement and expand a previous AIF neuromorphic study in a robotic system. For example, recreate Bayesian neurons [61] or Bayesian circuits [62] in neuromorphic hardware situated within a humanoid robot.

“One-shot” learning is also of particular interest. [34] used AIF to model concept learning and demonstrated “one-shot” generalisation to new stimuli. Although this study did not incorporate either robotics or neuromorphic hardware, it may be possible to implement a similar state–space expansion and reduction approach in a neurorobotic system (i.e., through exploring the role of “open slots”). This could also be combined with “Bayesian One-Shot learning” techniques to optimise multimodality training (e.g., [77]).

### 5.2. Benchmarking

The field would likely benefit from standardised benchmarking tasks. The OpenAI Gym platform (gymnasium.farama.org) has been valuable in providing this for the AI community, including AIF studies [78]. However, developmental neurorobotics (the focus of this perspective) requires embodied tasks due to the emphasis on self-determined learning via interaction with the environment. This field would additionally benefit from tasks capable of identifying trajectories and “stages” of learning over time, in addition to “mental age” mapping, in order to enhance the validity of comparisons with human development.

We, therefore, suggest that it may be valuable to adapt a battery of cognitive tests commonly used in children for use in embodied humanoid agents. For example, theory of mind tasks (such as the “Sally-Anne” test [79]), the Bayley scale for infant and toddler development [80], and specific tasks associated with each stage of Piaget’s stages of cognitive development (including tasks related to object permanence, conservation, classification, and abstract reasoning).

### 5.3. Open-Source Resources

The AI field has greatly benefited from open-source resources. We advocate for the development of a resource offering a range of different open-source biologically plausible neuromorphic circuits—such as cortical column, ring attractor, hippocampal, and central pattern generator. An existing open neuromorphic platform could be leveraged to create this—such as BrainScales2 [81], which offers spiking information transmission and plastic synapses. The development of such a resource would provide AI communities with modular composable blocks for creating novel neuromorphic solutions.

## Figures and Tables

**Table 1 entropy-26-00582-t001:** Summary of requirements for embodied “intelligent” agents (Sources: [15,29,39,40,41,42,43]).

Domain	Requirements
Cognition	Predictive, flexible, brain–body–environment model, incorporating value systems.
Computation	Sparse representations, efficient, morphological, incorporating information-processing biases.
Learning	Continual, open-ended, hierarchical multi-task, generalisable (including zero-shot), leveraging plasticity mechanisms.

**Table 2 entropy-26-00582-t002:** Summary of AIF implementation methods used in robotic studies.

Robot Skill (Relevant Studies)	Summary of Method for Implementing AIF
Body estimation [53,55]	Lanillos & Cheng [53] present body estimation and control as a free-model active inference problem combining state-of-the-art regressors with free energy lower bound minimization. Oliver et al. [55] approximated the robot’s body through variational inference (i.e., minimising the variational free energy (VFE) bound using the error between the expected and the observed sensory information). Free energy optimization was performed using gradient descent.
Body perception and action [71]	Used “PixelAI”, a novel pixel-based deep active inference algorithm, which provides model-free learning and unifies perception and action into a single variational inference formulation. It combines the FEP with deep convolutional decoders.
Navigation [56,57,72]	Çatal et al. [56] cast the simultaneous localization and mapping (SLAM) problem in terms of a hierarchical Bayesian generative model. The agent reasons on two different levels: on a higher level for long-term navigation and on a lower level for short-term perception. At the lower level, the model builds upon earlier work [26] (build and learn the generative model using deep artificial neural networks, which are trained on sequences of action–observation pairs), in order to learn and infer belief states from sequences of observations and actions. Burghardt & Lanillos [57] used laser sensors. They defined the true state (position) of the robot, and the position belief of the robot. Estimation was solved by computing the posterior distribution by optimizing VFE. Taniguchi et al. [72] combined sequential Bayesian inference using particle filters and information gain-based destination determination in a probabilistic generative model (“SpCoAE” method). The method achieves AIF by selecting candidate points with the maximum information gain and performing online learning from observations obtained at the destination.
Planning [73]	This study specified nominal behaviour offline through behaviour trees and used a “leaf node” to specify the desired state to be achieved, rather than an action to execute. The decision of which action to execute to reach the desired state was performed online through active inference, resulting in “continual online planning and hierarchical deliberation”.
Goal-directed behaviour [74]	This study brings together AIF-inspired behaviour generation and biologically plausible SNNs. It demonstrated that “goal-directed, anticipatory behaviour can emerge from projecting intentions through continuously unfolding spike dynamics onto motor input”.
Complex social cognition [59]	This study used a multi-layered “PV-RNN” model (this is an RNN-type model that can instantiate predictive coding and active inference in a continuous spatio-temporal domain) with two branches (vision and proprioception) connected through an associative module. The model predicts incoming visual sensation and proprioception simultaneously; prediction error is back-propagated through time, and each module is modulated to maximize the evidence lower bound.

## Data Availability

No new data were created or analyzed in this study. Data sharing is not applicable to this article.

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
