# Peer review of "Active Inference for Learning and Development in Embodied Neuromorphic Agents"

_entropy, 2024, doi:10.3390/e26070582_

Round 1
Reviewer 1 Report
Comments and Suggestions for Authors
The current paper claims that the Active Inference Framework (AIF) based on the Free Energy Principle (FEP) can enhance the learning and development of embodied brain-inspired artificial agents. The paper proposes that brain-like computation is inherently embodied and that learning, inference and generation processes with this embodiment requires operating a unified framework for autonomous mental development such as active inference. The paper presents the concepts clearly and precisely, making nontrivial ideas accessible to general readers. The comprehensive review of related literature and current state-of-the-art methods provides valuable context and highlights the novelty of the proposed framework. Although the paper is well written and worthwhile publishing in the current journal, the authors should address the following comments before the publication.
(1) In L227 it is written as
“Conceptually, AIF could potentially generate high-order cognitive and metacognitive capabilities, such as monitoring, self explainability and to some degree “awareness” (Lanillos et al., 2021).
Please explain how AIF can provide capabilities for monitoring, self-explanability, and awareness.
(2) In L239 it is written as
“AIF has been shown to perform as well as traditional ML methods in simple environments, and better in environments featuring volatility, ambiguity and context sensitivity.”
Although AIF could have such potential, I’m not sure if we can argue this strongly since only limited studies have been conducted for evaluating this aspect,. The authors may rewrite the expression.
(3) In L264 it is written as
“It was demonstrated that this approach naturally yields Hebbian plasticity. The authors suggest that this approach may dramatically reduce the complexity of designing self-learning neuromorphic hardware.”
I’m not sure why yielding Hebbian plasticity can lead to dramatic reduction of the complexity of designing self-learning neuromorphic hardware from this sentence. Please clarify this part.
(4) In L273 it is written as
“Importantly, capabilities emerged entirely from connectivity and from the “Bayesian engine” in neurons (rather than from hardwired learning rules).”
What is the Bayesian engine stored in neuron? Please explain this part.
(5) In L287 it is written as
“Note, the term “embodiment” here refers to the “separation of internal versus external states where feedback of the effect of an action on a given environment is available” (Kagan et al., 2022).”
Please clarify this sentence.
Author Response
Dear Reviewer,
Thank you very much for taking the time to review our manuscript and for providing helpful suggestions. Please refer to the tracked changes in the amended manuscript to review our revisions described here.
1. L227: “Conceptually, AIF could potentially generate high-order cognitive and metacognitive capabilities, such as monitoring, self explainability and to some degree “awareness” (Lanillos et al., 2021). Please explain how AIF can provide capabilities for monitoring, self-explanability, and awareness.
Thank you for highlighting this issue. We have added further detail from the Lanillos et al., 2021 paper to clarify what is meant by this in this context – particularly that agents are able to monitor their prediction precision.
2. L239: it is written as “AIF has been shown to perform as well as traditional ML methods in simple environments, and better in environments featuring volatility, ambiguity and context sensitivity.” Although AIF could have such potential, I’m not sure if we can argue this strongly since only limited studies have been conducted for evaluating this aspect,. The authors may rewrite the expression.
Thank you for this suggestion. We have amended the language in this section to argue this point less strongly.
3. L264 it is written as “It was demonstrated that this approach naturally yields Hebbian plasticity. The authors suggest that this approach may dramatically reduce the complexity of designing self-learning neuromorphic hardware.” I’m not sure why yielding Hebbian plasticity can lead to dramatic reduction of the complexity of designing self-learning neuromorphic hardware from this sentence. Please clarify this part.
Thank you for highlighting this particular issue. We have amended this to clarify further the findings of the authors (delayed modulation of Hebbian plasticity, which is important for delayed reward tasks, is a realisation of active inference). The authors claim the implication for this is low computation cost and a simple architecture in designing neuromorphic hardware. We have amended this paragraph to clarify these author observations.
4. L273: it is written as “Importantly, capabilities emerged entirely from connectivity and from the “Bayesian engine” in neurons (rather than from hardwired learning rules).” What is the Bayesian engine stored in neuron? Please explain this part.
We have amended the sentence to clarify this is in relation to individual neurons acting to minimise their prediction error. Thank you for this suggestion.
5. L287 it is written as: “Note, the term “embodiment” here refers to the “separation of internal versus external states where feedback of the effect of an action on a given environment is available” (Kagan et al., 2022).” Please clarify this sentence.
This particular sentence is a quote from the Kagan et al study, where they are defining what they mean by embodiment in the context of their experiment. We have therefore added a further sentence to put this in the context of perspectives on embodiment and directed to a paper discussing this further (Manzoitti). Thank you for highlighting this.

Reviewer 2 Report
Comments and Suggestions for Authors
See attached file

Author Response
Dear reviewer,
Thank you very much for taking the time to review our manuscript and for providing helpful suggestions. Please refer to the track changes in the amended manuscript to review our revisions described here.
We very much appreciate you highlighting that the review would benefit from extending what AIF is in more detail, to improve accessibility. We have therefore created a summary/explainer box to highlight key concepts, benefits and challenges associated with active inference. Please note there is a comparison with reinforcement learning provided (line 150).
“It is in general unclear to me that embodied intelligence is expected to emulate human behavior or learning. Embodied intelligent objects can achieve all kinds of tasks, many of which are not expected from humans on account of their physical limitations, which are not appropriate to generic robots. Therefore it is not clear to me what is meant by comparing the “mental age” of such agents to human development.”
Thank you for this observation. While we agree many capable agents are embodied, traditionally these solutions must be explicitly programmed, and are not adaptable to changes in the physical embodiment itself. Consequently, our review draws upon insights from the field of Developmental Neurorobotics. Embodiment is a key feature within this field because the approach emphasises ongoing self-determined learning via interaction with the environment, rather than pre-training. We emphasise Developmental Neurorobotics as a paradigm which could be leveraged to catalyse the creation of agents that are autonomous and adaptive, and suggest the AIF could be well suited to implement such processes.
We focus on human-like learning in this review, and therefore benchmarking physical humanoid implementations with human cognitive tests would be appropriate in this context. We agree in other physical implementations this would not make sense, and have therefore clarified this is in relation to mapping learning trajectories in humanoid implementations. Please see amendments to section 5.2. Thank you very much for highlighting this issue.
“In line 264, I cannot understand the sentence “It was demonstrated that, delaying this approach naturally yields Hebbian plasticity naturally yields active inference””
Thank you for identifying this issue. We have amended this paragraph to further clarify the findings and observations of this particular study – including the relevance of Hebbian plasticity.
What is “e-prop” algorithm? Please include more details, as you refer to it specifically.
The e-prop (eligibility propagation) learning algorithm is a method designed for training spiking neural networks (SNNs). It relies on local learning rules, meaning that the weight updates depend only on information available at each synapse. We have amended the sentence to summarise these details. Thank you for this suggestion.
Typo in “the” on line 346. Typo in “spiking” on line 371.
Thank you very much for identifying these. We have amended the manuscript to correct both typos.

Round 2
Reviewer 2 Report
Comments and Suggestions for Authors
The authors have addressed my main concerns in this revision. I am happy to recommend it for publication.